# Strength Prediction of Non-Sintered Hwangto-Substituted Concrete Using the Ultrasonic Velocity Method

**DOI:** 10.3390/ma17010174

**Published:** 2023-12-28

**Authors:** Hajun Im, Wonchang Kim, Hyeonggil Choi, Taegyu Lee

**Affiliations:** 1Department of Fire and Disaster Prevention, Semyung University, Jecheon 27136, Republic of Korea; haha1578@naver.com (H.I.); kimwc69082@gmail.com (W.K.); 2School of Architecture and Civil Engineering, Kyungpook National University, Daegu 41566, Republic of Korea

**Keywords:** non-sintered Hwangto, mechanical properties, ultrasonic pulse velocity, compressive strength, prediction model, regression analysis, microstructural analysis

## Abstract

This paper presents and investigates the properties of concrete in which a portion of the cement is substituted with non-sintered Hwangto (NSH), a readily available building material in Asia. Given the inactive nature of NSH, this study aimed to determine the optimal cement replacement ratio and quantitative strength of the material. The unit weight, compressive strength, ultrasonic pulse velocity (UPV), and stress–strain of the NSH concrete (NSHC) were evaluated. Additionally, we developed a predictive model for determining compressive strength based on the regression analysis of compressive strength and UPV. The water-to-binder ratio was set to 0.41, 0.33, and 0.28, and the NSH replacement rates in the cement were set to 0%, 15%, 30%, and 45% for evaluating various strength ranges. The mechanical property measurements indicated reductions of 5.35% in unit weight, 35.62% in compressive strength, and 6.34% in UPV as the NSH was replaced. Notably, the smallest deviation from plain concrete was observed at a replacement rate of 15%. The scanning electron microscopy analysis results showed that the plain concrete exhibited a crystalloid structure; however, as the NSH replacement rate increased, the amorphous structure and pores increased while unreacted NSH particles were also observed. The X-ray diffraction analysis results demonstrate that the peak intensities for kaolinite and mullite increased as the NSH replacement rate increased, while those of C–S–H gel and CaO showed low peak intensities. Furthermore, the regression analysis concluded that an exponential function was suitable. Consequently, a compressive strength prediction model was developed, and in the error test, the NSHC model demonstrated an average error of <10%, with fewer errors at the lower compressive strength boundaries.

## 1. Introduction

Following the announcement of “Net Zero Emission 2050” as a carbon neutral scenario by the International Energy Agency in 2022, the importance of achieving carbon neutrality has gained prominence globally. Additionally, since the 2015 Paris Agreement, several countries have been actively conducting research to curtail carbon emissions, especially in the construction sector, where the main focus has been to find alternatives to cement [1,2]. Cement—a ubiquitous construction material—considerably contributes to greenhouse gas emissions during its production process, substantially impacting climate change [3]. In 2017, emissions from cement manufacturing were reported at 36 billion tons, with carbon dioxide emissions constituting ~7% of the total greenhouse gas emissions. Given the expected increase in the annual cement demand, there is a pressing need for research into materials that can effectively substitute cement without compromising performance [4,5,6,7,8]. Accordingly, research is being conducted to reduce carbon emissions by using cement replacement materials or analyzing the LCA of concrete using replacement materials [9].

Current research encompasses the utilization of various industrial by-products and natural resources [10,11,12]. Hwangto, a material with a widespread geographical distribution covering approximately 10% of the Earth’s surface, offers the benefits of a stable supply chain [8]. As a clay pozzolan belonging to the meta-kaoline mineral series, its primary components are similar to those in fly ash and blast furnace slag, previously employed cement substitutes [13]. Prior studies have reported that using sintered Hwangto can improve the strength and durability of concrete or mortar [14]. However, substantial amounts of heat energy and greenhouse gases are emitted during the Hwangto sintering or activating process, posing environmental risks. Moreover, costs for constructing the required infrastructure for Hwangto need to be considered. Consequently, research in Asian regions is increasingly focused on utilizing non-sintered Hwangto (NSH) as a viable cement alternative. 

Figure 1 shows the non-sintered Hwangto that we used in this experiment. Previous studies have indicated that swelling due to water absorption by Hwangto can result in the formation of physical bonds between particles; however, when the test specimen was compacted at low pressure, it underwent large deformation due to pore collapse [15]. Consequently, ongoing research efforts are directed toward enhancing the surface characteristics of Hwangto particles and addressing their low mechanical properties caused by high adsorption rates by pores.

Despite these characteristics, as the NSH is fabricated without a sintering process, it delivers lower strength than concrete composed solely of cement. Concrete using NSH cannot be expected to develop strength due to pozzolanic reactions in the short or long term. Therefore, to effectively use NSH as an admixture to replace a portion of cement, continued research is essential for determining the optimal replacement rate, considering the mechanical properties of concrete. 

The utilization of non-sintered Hwangto concrete (NSHC) as a primary structural component in buildings requires a thorough evaluation of concrete strength development and precise strength prediction. Such assessments are crucial for ensuring the safety of structural elements during the construction, maintenance, and repair phases [16,17,18,19]. Traditionally, the compressive strength was assessed by extracting cores through drilling—a method that poses safety concerns. Among the non-destructive techniques employed for estimating compressive strength, the use of ultrasonic pulse velocity (UPV), which employs ultrasonic waves, has gained prominence. Consequently, numerous researchers are employing UPV to evaluate the mechanical properties or quality of concrete [20,21,22]. However, in the context of NSHC, additional research and analysis are required due to the lack of extensive experimentation.

In this study, in order to measure the mechanical properties of concrete containing Hwangto, a natural mineral, SEM and X-ray diffraction (XRD) analyses were performed by evaluating mechanical properties, such as the unit weight, compressive strength, and stress–strain, and the microstructure of NSHC concrete produced with various W/B and substitution ratios was measured. Furthermore, a prediction model was developed by analyzing the correlation between the UPV and compressive strength. Through regression analysis, a strength prediction equation was proposed for NSHC.

## 2. Experimental Procedure

### 2.1. Experimental Plan 

The experimental design of this study is outlined in Figure 2. The binder was defined as a combination of cement and NSH, with the NSH substitution ratios in the cement set at 0%, 15%, 30%, and 45%. Additionally, the water-to-binder (W/B) ratios were established at 0.41, 0.33, and 0.28 to assess various target compressive strengths (30, 45, and 60 MPa). The measurements were categorized into three primary areas: mechanical properties, microanalysis, and statistical analysis. Mechanical property assessments included unit mass, compressive strength, UPV, and stress–strain rate to evaluate both strength and stiffness. Microanalysis involved XRD and SEM to analyze the composition and microstructure of the specimens. Statistical analysis included the development of a strength prediction model for NSHC through regression analysis correlating UPV with compressive strength, complemented by an error assessment through comparison with prior studies. Figure 3 shows the NSHC preparation overview of this study.

### 2.2. Materials

The physical properties of the materials utilized in this research are summarized in Table 1. The cement used was type Ⅰ ordinary Portland cement (OPC), characterized by a density of 3150 kg/m^3^ and fineness of 320 m^2^/kg. NSH, employed as a cement substitute, exhibited a density of 2500 kg/m^3^ and fineness of 330 m^2^/kg. The coarse aggregate was crushed granite aggregate with a density of 2680 kg/m^3^, a fineness modulus of 7.03, an absorption rate of 0.68%, and a maximum size of 20 mm. The fine aggregate, river sand, had a density of 2540 kg/m^3^, a fineness modulus of 2.54, and an absorption rate of 1.6%.

The chemical properties of OPC and NSH are detailed in Table 2. NSH, a kaolin-based mineral, exhibited lower CaO content but higher concentrations of SiO_2_ and Al_2_O_3_ compared to cement. The NSH particles are characterized by a rough surface, which has been associated with low strength development [23]. Furthermore, NSH exhibits strong hydrophilicity and interparticle condensation, leading to volume changes in response to moisture. Although these properties suggest its potential applicability as a building material, the low density and high porosity of NSH result in reduced strength and increased deformation when used as a cement substitute [24]. The chemical composition of kaolin-based clay materials can vary by region. The NSH used in this study was sourced from Jeollanam-do, Korea.

### 2.3. Mix Proportion, Casting, and Curing of Specimens

The mix proportions for this study are listed in Table 3. The concrete specimens were categorized into plain (not containing NSH) and NSHC (containing NSH). In the “MIX ID”, the numbers following plain and NSHC indicate the W/B ratios, while “−15”, “−30”, and “−45” represent the NSH substitution rates for cement. Owing to the hydrophilic nature and strong condensation force of NSH in the presence of moisture, a specific mixing sequence was employed: cement and NSH were initially mixed, followed by the sequential addition of aggregate and water, to ensure uniform distribution of NSH particles.

The specimens were fabricated as cylindrical models measuring Φ 100 × 200 mm, with the curing periods set to 1, 3, 7, and 28 days. Curing involved maintaining a constant temperature and humidity until the first day of aging, thereafter transitioning to underwater curing until the 28th day of aging.

### 2.4. Test Method

The mechanical property analysis conducted in this study is outlined in Table 4. The mechanical properties were assessed at 1, 3, 7, and 28 days, with values calculated by averaging the three test specimens. Compressive strength was measured in accordance with ASTM C39/39M, and strain–stress was evaluated based on ASTM C469 [25,26]. Figure 4 illustrates the UPV measurements conducted following ASTM C597, with UPV being calculated by dividing the travel time by the distance [27]. Prior to the UPV measurement, vacuum grease was applied to the transducer to ensure optimal contact between the transducer and concrete surface at both ends. Table 5 shows the quality of concrete according to the UPV range.

This study conducted a regression analysis of compressive strength and UPV relative to the W/B ratio and NSH substitution rate. This analysis facilitated the development of a strength prediction model for plain concrete and NSHC using UPV analysis. An error test was performed to compare the findings with that of previous studies and establish standards, thereby verifying the error rate in strength prediction.

## 3. Results and Discussion

### 3.1. Mechanical Properties on Plain Concrete and NSHC by Age

#### 3.1.1. Unit Weight

Figure 5 illustrates the unit weight of plain concrete and NSHC at varying ages with an NSH substitution rate of 45%. When comparing concretes with identical W/B ratios, the unit weight decreased with an increase in the NSH substitution rate. At 28 days, Plain41 showed a unit weight of approximately 2280 kg/m^3^, while NSHC41-45 exhibited 2115 kg/m^3^, with a tendency to decrease by ~7.23%. For Plain33, the unit weight was approximately 2313 kg/m^3^, and for NSHC33-45, it was 2182 kg/m^3^, representing a decrease of ~5.67% compared to Plain33. The highest density within the plain series was observed for Plain28 (2434 kg/m^3^), while NSHC28-45 registered a unit weight of 2357 kg/m^3^, a reduction of 3.14% compared to plain concrete. This trend suggested that a higher compressive strength of W/B corresponds to a less pronounced difference in unit weight. The reason for the difference in unit weight is believed to be due to the difference in materials in the experimental mixing.

#### 3.1.2. Compressive Strength

Figure 6 presents the compressive strength of plain concrete and NSHC over time. Generally, an increase in the NSH substitution rate corresponded to lower compressive strength. On day 1 of aging, Plain41 exhibited a compressive strength of ~9 MPa. Meanwhile, NSHC41-15, NSHC41-30, and NSHC41-45 exhibited strengths of approximately 6.58, 4.60, and 2.16 MPa, respectively. For Plain33, the strength was ~16.69 MPa, while NSHC33-15, NSHC33-30, and NSHC33-45 displayed strengths of 14.66, 10.87, and 5.46 MPa, respectively. Plain28 exhibited a strength of ~35.08 MPa, with NSHC28-15, NSHC28-30, and NSHC28-45 displaying strengths of 23.73, 16.72, and 11.91 MPa, respectively. The smaller the W/B of the concrete mix, the higher the compressive strength of the concrete, and as the NSH substitution ratio increased, the compressive strength decreased. The relatively minor strength differences at day 1 of aging compared to plain concrete are attributed to temporary shrinkage at the initial age due to the high absorbability of NSH, resulting in reduced cohesion [28]. Additionally, previous research has indicated that the hydrophilic nature of Hwangto aids in absorbing natural and surface water, which are then utilized in the cement hydration reaction, resulting in diminished hydrate formation [29,30].

At 28 days, plain concrete exhibited compressive strengths of 34.65, 45.60, and 61.30 MPa, surpassing the respective target strengths. Replacing 15% of cement with NSH caused a marginal reduction in the compressive strength, and the values remained close to the target strengths (NSHC41-15: 27.44 MPa, NSHC33-15: 41.29 MPa, and NSHC28-15: 51.37 MPa). Prior research involving fly ash, blast furnace slag, and other kaoline-based admixtures as cement substitutes suggested an optimal replacement rate of ~5 to 15%. Beyond 20% substitution, significant mechanical property degradation was observed [31,32,33]. In this study, a 30% NSH substitution manifested in reduced compressive strength (NSHC41-30: 21.53 MPa, NSHC33-30: 32.42 MPa, and NSHC28-30: 43.03 MPa). With 45% cement replacement by NSH, notably low strength development was noted (NSHC41-45: 11.12 MPa, NSHC33-45: 15.07 MPa, and NSHC28-45: 30.12 MPa).

Previous studies incorporating NSH in concrete recorded a decline in compressive strength with increasing NSH rates. Sung et al. (2022) identified a decrease in compressive strength attributable to the increased NSH substitution rate, highlighting the reduction in cement, a critical element in strength development [34]. Cho et al. (2008) observed that NSH, being a porous material with an open matrix structure, low specific gravity, and small particles, negatively impacted the strength owing to its high porosity and consequent moisture requirements, resulting in shrinkage [35]. Consequently, an optimal NSH substitution rate should be determined. This study concludes that a substitution rate of approximately 15% is appropriate.

#### 3.1.3. Ultrasonic Pulse Velocity

Figure 7 displays the UPV of concrete at various ages for different W/B ratios. Generally, the UPV increased with age, and a lower W/B ratio correlated with higher UPV. This increase in UPV over time and with decreasing W/B, as reported in prior studies, is attributed to the mixed unit cement amount and the reduced porosity and increased hydration products in the concrete matrix at lower W/B ratios [36,37].

Parallel to the pattern observed in compressive strength, an increase in the NSH substitution ratio decreased the UPV. On day 1 of aging, the UPV of Plain41, Plain33, Plain28 was 3.07, 3.42, and 4.17 km/s, respectively. However, the NSHC specimens with W/B ratios of 0.41 and 0.33 recorded UPVs of less than 3 km/s. Sung et al. (2003) reported that a decrease in UPV depends on the NSH substitution rate, with specimens containing NSH up to 7 days of aging exhibiting UPVs of less than 3 km/s [38]. UPV is considerably influenced by the physical properties of the material, such as porosity and microcracks. A lower UPV in NSH mixtures is believed to result from its smaller particle size compared to cement, higher porosity, and propensity to crack due to shrinkage.

At 28 days, the plain and NSH test specimens with a replacement rate of 15% exhibited UPVs of more than 4 km/s. NSHC41-30 displayed a UPV of 3.82 km/s, while that for NSHC33-30 and NSHC28-30 reached above 4 km/s. Meanwhile, with a 45% NSH substitution rate, only NSHC28-45 demonstrated a UPV exceeding 4 km/s. According to Table 6 which outlines the concrete quality standards based on the UPV range, all specimens except NSHC41-30, NSHC41-45, and NSHC33-45 were classified as “Good” or higher [39]. In terms of compressive strength, increasing NSH substitution decreased the compressive strength to approximately 67.89% compared to plain concrete; however, as the NSH substitution rate increased, UPV demonstrated a lower UPV of up to 16.55% compared to plain concrete, displaying a small difference compared to the compressive strength.

#### 3.1.4. Stress–Strain Curve

Figure 8 presents the stress–strain curve on plain concrete and NSHC by age. Until day 1, all specimens exhibited similar stress–strain curves, but notable differences emerged from day 3. Plain41 demonstrated a peak strain of 0.002130, which is similar to that displayed by NSHC41-30 (0.002012). For an NSH replacement rate of 45%, very low peak strain was observed, with the slope of the curve decreasing relatively consistently as the NSH substitution rate increased. The stress–strain curves for Plain33 and NSHC33-15 exhibited similar patterns, but specimens with a 45% NSH substitution rate demonstrated peak strains less than half of that observed at other substitution rates.

For specimens with a W/B ratio of 28, characterized by high target compressive strength, Plain28 and NSHC28-15 displayed relatively similar curves. NSHC28-30 exhibited the highest peak strain of 0.002820, while NSHC28-45 exceeded the slope of NSHC28-30 but recorded the smallest strain of 0.001648 in the same W/B range. Previous research has indicated that concrete in the high-strength range typically exhibits relatively brittle failure [40]. Consistent with this finding, this study observed that the peak strain in the W/B 28 range was lower than that observed at other W/B ratios.

### 3.2. Microstructural Analysis of Plain Concrete and NSHC

#### 3.2.1. SEM

Figure 9 illustrates the results of the SEM analysis on W/B 0.41 plain concrete and NSHC. Natural minerals like NSH, characterized by their relatively rough surfaces and hydrophilic nature, can absorb moisture, potentially impeding the hydration reaction. Figure 7a depicts the SEM results for plain concrete, where a denser structure was observed, attributed to the properties of calcium hydroxide (Ca(OH)_2_) and calcium silicate hydrate (C–S–H) gel. Figure 9b–d present the SEM results of concrete with NSH substitution. With increasing NSH rates, a proliferation of pores and an amorphous matrix structure were evident in NSHC. However, at a 15% NSH substitution level, the matrix appeared relatively dense compared to specimens with 30% and 45% substitution, with it bearing similarity to the structure of plain concrete. Additionally, higher NSH substitution rates led to the formation of smaller hydration product particles due to reduced hydration reactions.

#### 3.2.2. XRD

Table 6 shows the density and elastic properties of molecules, including the density, bulk modulus, and Poisson’s ratio of each molecule. Figure 10 displays the results of the XRD analysis based on the NSH replacement ratio. The XRD analysis focused on a specimen with a W/B ratio of 0.41 at 28 days of aging. Hwangto is a kaoline-based soil mineral, and its constituent materials include kaolinite (Al_2_Si_2_O_5_(OH)_4_), mullite (3Al_2_O_3_·2SiO_2_), and illite [41]. As the present XRD results pertained to mortar, materials with low intensity peaks compared to the crystal structure of cement-based substances were not displayed; here, only kaolinite and mullite exhibited high-intensity peaks. An increased intensity was noted for kaolinite’s peaks at higher NSH substitution rates [42]. The presence of kaolinite in the plain concrete is expected to be influenced by the fine aggregate used in the mortar mix. Similarly, the peak intensity of mullite, another principal mineral in Hwangto, rose with increasing NSH substitution. In contrast, the peak intensities for calcium hydroxide (Ca(OH)_2_) decreased as the NSH substitution rate increased. The C–S–H gel also showed lower peak intensity with higher NSH content. Given the inert nature of NSH, limited hydration reactions are expected from pozzolanic activity, suggesting that the peaks related to the main crystal structure are more pronounced at lower NSH substitution rates. 

### 3.3. Result of Regression Analysis

#### 3.3.1. Regression Analysis of Plain Concrete and NSHC

Figure 11 presents the strength prediction model for plain concrete and NSHC for various W/B ratios and NSH replacement rates. Existing strength prediction models typically feature exponential, linear, and quadratic functions, depending on the data distribution [43,44,45,46,47,48,49,50,51,52,53]. Table 7 shows the reliability analysis of regression models. In this study, the exponential function model was deemed the most suitable. All prediction models indicated a rapid increase in compressive strength in the higher UPV ranges. Although this observation suggests that UPV increases with compressive strength, the variations in UPV were not significant beyond a certain high-compressive-strength range. Previous research on strength prediction models incorporating existing cement replacement materials and UPV has analyzed plain concrete data. 

However, plain concrete typically exhibited higher compressive strength compared to concrete with substitute materials, particularly those using natural minerals that have not undergone a firing process, resulting in lower mechanical properties. Consequently, this study posits the necessity for a compressive strength prediction model specifically tailored to NSHC.

#### 3.3.2. Error Test with Previous Research

Figure 12 compares the new prediction model with prior research for various W/B ratios. Figure 12a compares the existing prediction model with the NSHC model of this study. Here, the line in the legend represents the curve of the existing prediction model. The NSHC prediction equation was ${y=0.1552e}^{1.2851x}$ and the correlation coefficient (R^2^) was 0.9331. Compared to previous models, such as that of Lee et al. [15], the NSHC model exhibited a steeper slope; however, both models were mostly similar. Figure 12b illustrates a comparison between experimental and predicted data from previous studies and NSHC. The graph denotes a 10% error increase from the inner to the outer line, with the trend lines of each dataset represented as dotted lines in a linear function format. For NSHC, errors were less than 10% at lower strengths but were approximately 18% at higher compressive strengths. The error trends of the NSHC and AIJ (Architectural Institute of Japan) models were mostly similar. Across all studies, lower errors were observed for strengths below 30 MPa.

## 4. Conclusions

In this study, we assessed the mechanical properties of concrete where a substantial amount of cement was replaced with NSH, developed a compressive strength prediction model using UPV, and summarized the findings as follows:

Although the plain specimen achieved the target compressive strength at all W/B ratios, NSHC did not achieve the target strength in all cases. By the 28th day, NSHC41-15 demonstrated only a decrease of approximately 8.5% in strength compared to plain concrete, while NSHC33-15 displayed an about 8.2% decrease and NSHC28-15 showed an approximately 14.4% decrease. These results suggest that an optimal NSH substitution rate is approximately 15%.UPV decreased with increasing NSH substitution ratios. However, unlike compressive strength, the difference ratio between NSHC and plain concrete was not significant. At 28 days, all specimens except NSHC41-30, NSHC41-45, and NSHC33-45 exhibited UPV speeds above 4 km/s. According to concrete quality standards based on the UPV range, these are rated as “Good” or higher. Because UPV is propagated through molecular vibrations, it can be inferred that NSHC, containing molecules with a relatively low bulk modulus, exhibited lower UPV than the plain concrete.The stress–strain curves displayed similar patterns at the initial phase of aging but demonstrated changes after 3 days. Concrete with high compressive strength displayed low peak strain, indicating a more brittle nature, with the steepest slope curve observed in specimens with high compressive strength W/B ratios.SEM analysis revealed that with increasing NSH substitution rates, the number of voids increased and an amorphous matrix structure became more prevalent, interspersed with NSH particles. This structural change contributed to lower mechanical properties compared to the plain concrete. Furthermore, XRD analysis showed that the intensity of kaolinite and mullite peaks increased with the increasing NSH substitution rate, while that of C–S–H gel, Ca(OH)2, and CaO decreased as the NSH substitution rate increased.The regression analysis of UPV and compressive strength indicated that an exponential function is the most suitable for the equation predicting the compressive strength for all specimens. Beyond a certain UPV threshold, the compressive strength exhibited a steep increase. Although UPV increases with compressive strength, the rate of UPV increase diminishes for a higher compressive-strength value. The error testing between the predicted and experimental values demonstrated that the lower the compressive strength, the smaller the error. On average, NSHC exhibited an error of less than 10%.It was confirmed that the smaller the substitution rate of NSH, the similar the mechanical properties to plain concrete. However, we believe that research on concrete using cement substitutes to reduce carbon should continue. Accordingly, research should be conducted on wide W/B and detailed cement replacement ratios using various materials, including NSH.

## Figures and Tables

**Figure 1 materials-17-00174-f001:**
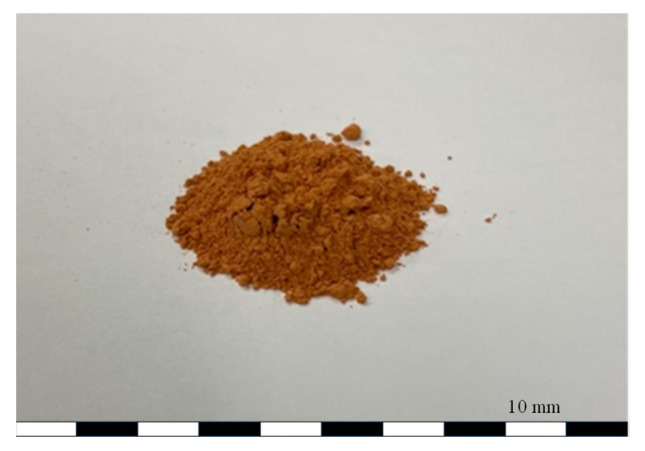
Non-sintered Hwangto (NSH).

**Figure 2 materials-17-00174-f002:**
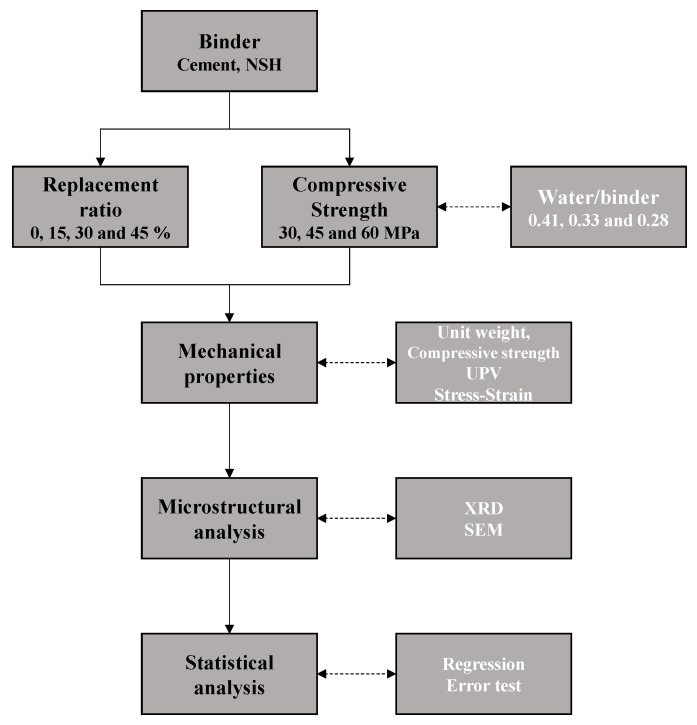
Workflow diagram of experimental plan.

**Figure 3 materials-17-00174-f003:**
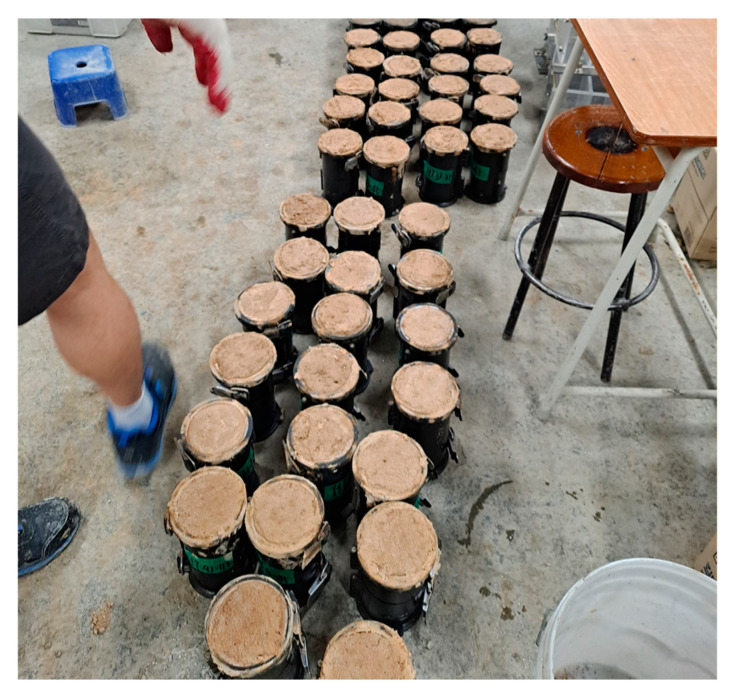
NSHC preparation overview.

**Figure 4 materials-17-00174-f004:**
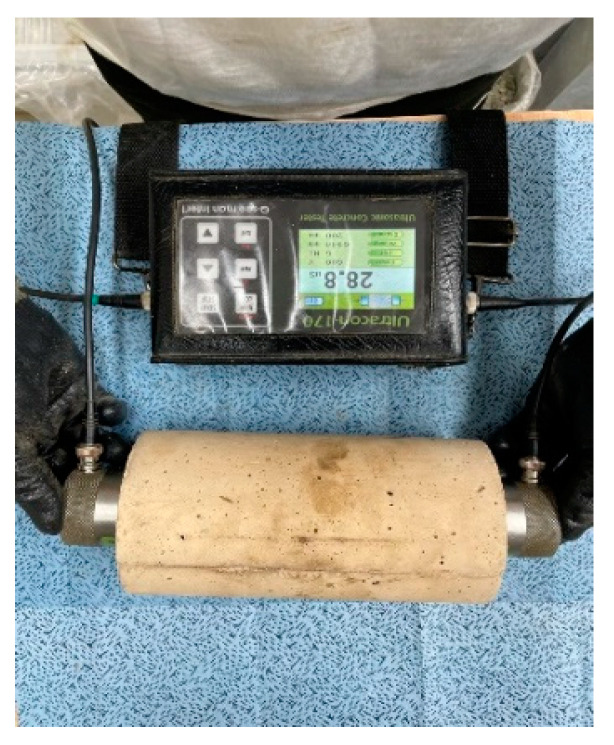
Ultrasonic pulse velocity (UPV) test.

**Figure 5 materials-17-00174-f005:**
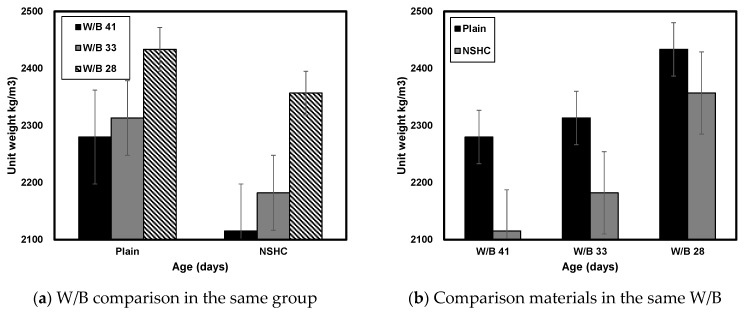
Comparison of the unit weight of concretes at an NSH substitution rate of 45%.

**Figure 6 materials-17-00174-f006:**
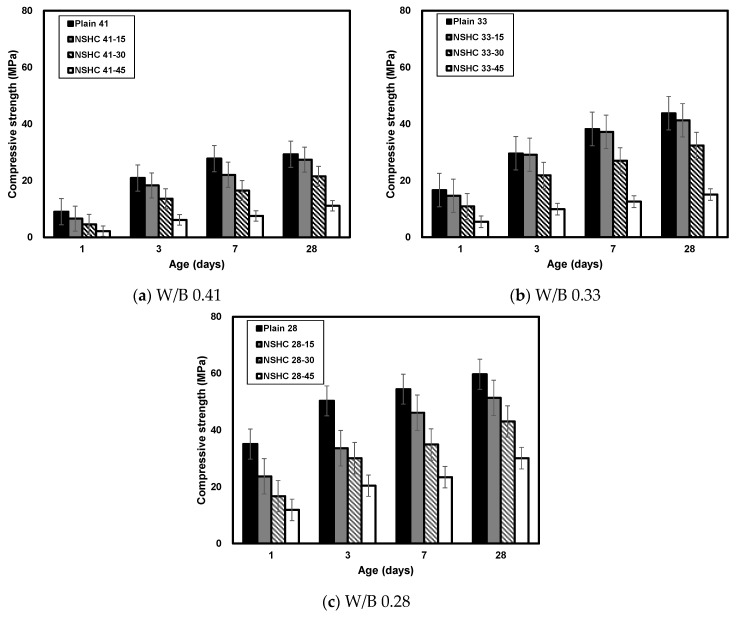
Comparison of the compressive strength of concrete by age at various water/binder (W/B) ratios.

**Figure 7 materials-17-00174-f007:**
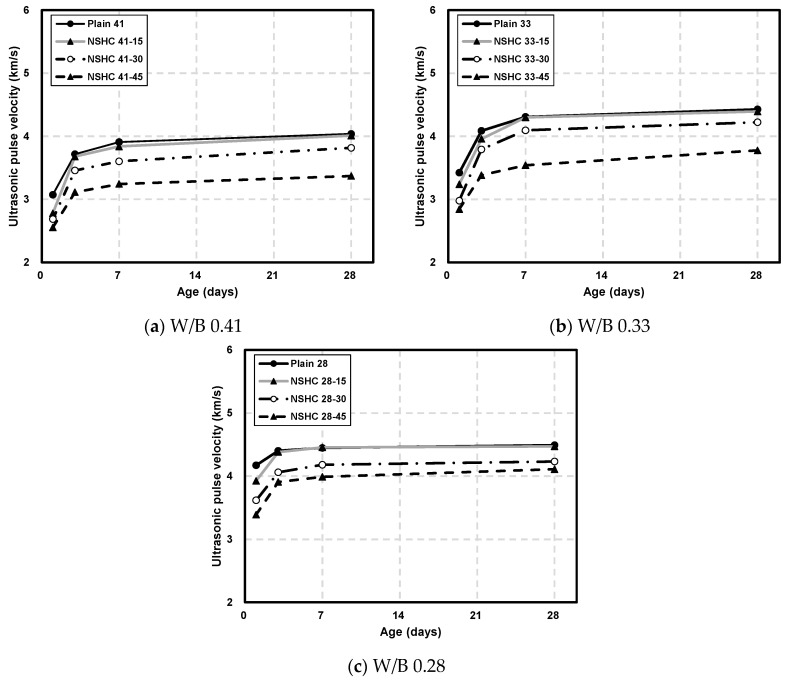
Comparison of the UPV of concrete by age at various W/B ratios.

**Figure 8 materials-17-00174-f008:**
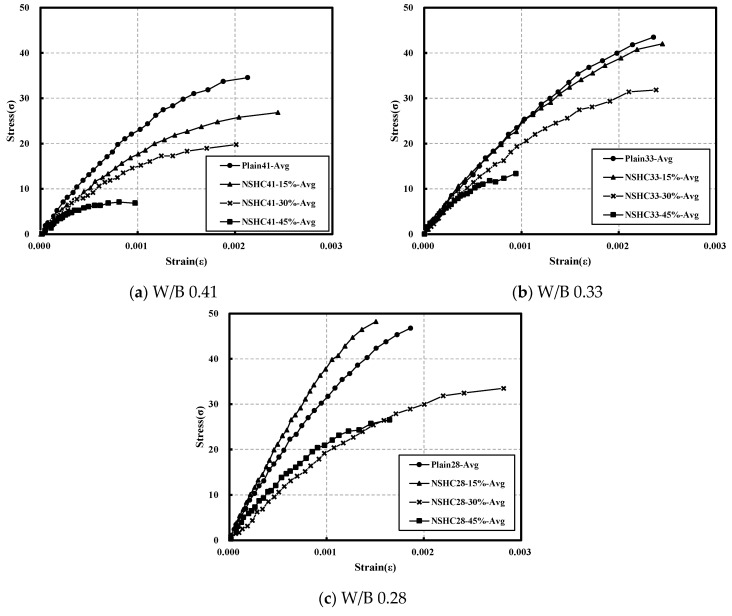
Comparison of average stress–strain curves for plain concrete and NSHC at various W/B ratios.

**Figure 9 materials-17-00174-f009:**
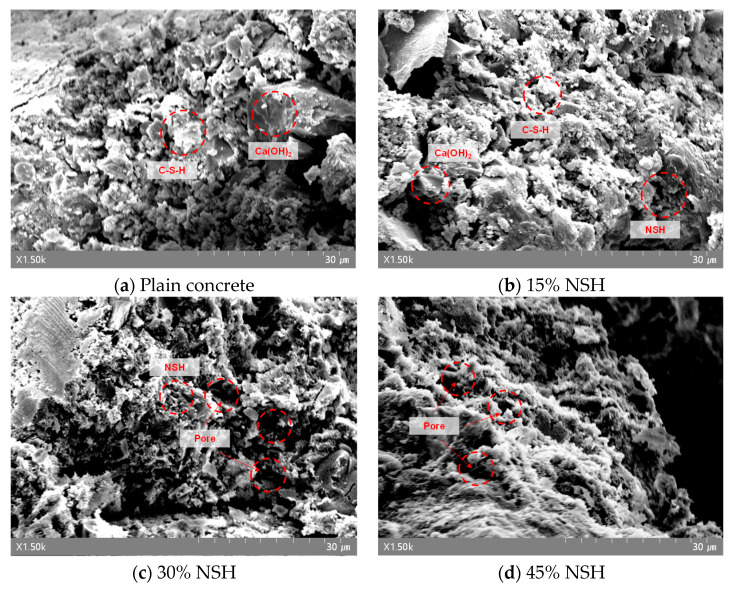
Scanning electron microscope analysis results of plain concrete and NSHC according to the NSH substitution ratio of W/B 0.41 concrete.

**Figure 10 materials-17-00174-f010:**
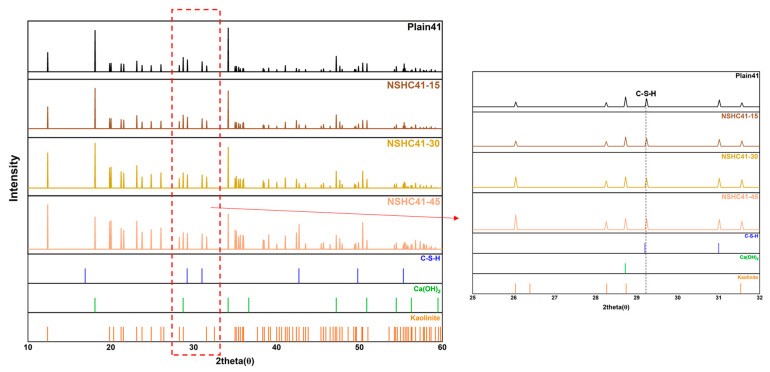
Results of X-ray diffraction analysis according to the NSH replacement ratio.

**Figure 11 materials-17-00174-f011:**
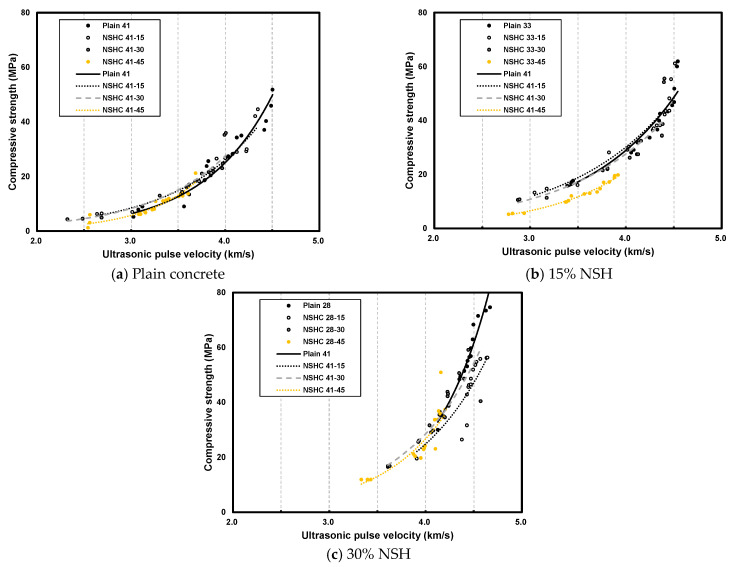
Prediction model for plain concrete and NSHC at various W/B ratios and NSH replacement rates.

**Figure 12 materials-17-00174-f012:**
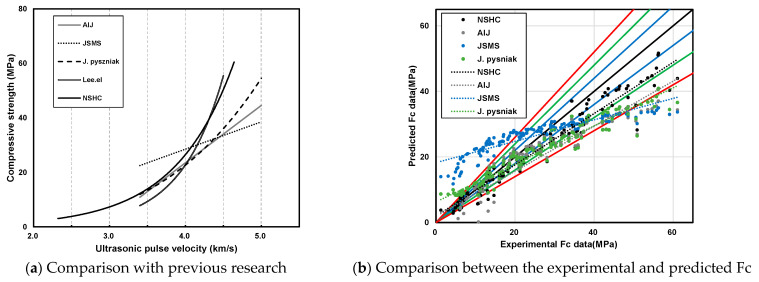
Comparison of the proposed prediction model with that reported in previous research at various W/B ratios.

**Table 1 materials-17-00174-t001:** Physical properties of the materials.

Materials	Properties
Cement	Type Ⅰ ordinary Portland cement Density: 3150 kg/m^3^ and fineness: 320 m^2^/kg
Hwangto	Non-sintered Hwangto Density: 2500 kg/m^3^ and fineness: 330 m^2^/kg
Coarse aggregate	Crushed granite aggregate Density: 2680 kg/m^3^ and fineness modulus: 7.03, Absorption: 0.68% and maximum size: 20 mm
Fine Aggregate	river sand Density: 2540 kg/m^3^, fineness modulus: 2.54, and absorption: 1.6%

**Table 2 materials-17-00174-t002:** Chemical properties of OPC and NSH.

Materials	Chemical Composition (%)								L.O.I.
	CaO	SiO_2_	Al_2_O_3_	Fe_2_O_3_	MgO	SO_3_	K_2_O	Others	
OPC	60.34	19.82	4.85	3.30	3.83	2.88	1.08	0.86	3.02
NSH	0.39	40.0	32.9	7.79	1.54	–	0.76	16.62	13.7

OPC: ordinary Portland cement; NSH: non-sintered Hwangto; L.O.I: loss on ignition.

**Table 3 materials-17-00174-t003:** Mix proportions of the plain and NSH concrete (NSHC).

MIX ID	W/B	S/a	Unit Weight (kg/m^3^)				
			W	C	NSH	S	G
Plain41	0.41	0.46	165	400	–	799	956
NSHC41-15				340	60	794	950
NSHC41-30				280	120	788	943
NSHC41-45				220	180	781	935
Plain33	0.33	0.43		500	–	711	961
NSHC33-15				425	75	705	953
NSHC33-30				350	150	699	944
NSHC33-45				275	225	691	934
Plain28	0.28	0.43		600	–	676	896
NSHC28-15				510	90	668	886
NSHC28-30				420	180	661	876
NSHC28-45				330	270	655	866

W/B: water/binder; S/a: sand/aggregate; W: water; C: cement; NSH: non-sintered Hwangto; S: sand; G: gravel.

**Table 4 materials-17-00174-t004:** Mechanical property analysis.

Materials	Properties	Equation (1)
Compressive strength (MPa)	ASTM C39/C39M	$v_{p}=\frac{L}{t}$ *v_p_*: ultrasonic pulse velocity (m/s) L: distance (m) t: time (s)
Strain (σ)–stress (ε)	ASTM C469	
Ultrasonic pulse velocity (km/s)	ASTM C597	

**Table 5 materials-17-00174-t005:** Quality of concrete according to the UPV range.

Range of UPV	Quality of Concrete
Exceeding 4500 m/s	Excellent
4000–4500 m/s	Good
3500–4000 m/s	Doubtful/questionable
3000–3500 m/s	Poor
Less than 3000 m/s	Very poor

**Table 6 materials-17-00174-t006:** Density and elastic properties of molecules.

Properties	Chemical Composition (%)						
	SiO_2_	Al_2_O_3_	Fe_2_O_3_	CaO	MgO	SO_3_	K_2_O
Density (kg/m^3^)	2630	3950	5190	3350	3630	2340	2410
Bulk modulus (GPa)	29	231	178	105	151	5	27
Poisson’s ratio	0.00	0.24	0.31	0.21	0.19	0.31	0.27

**Table 7 materials-17-00174-t007:** Reliability analysis of the regression models.

ID	Exponential			Linear		
	R-Square	Root-MSE	RMS	R-Square	Root-MSE	RMS
Plain41	0.92	4.07	16.55	0.91	4.46	19.85
NSHC41-15	0.88	3.90	15.22	0.83	4.69	22.01
NSHC41-30	0.80	4.26	18.19	0.77	4.69	21.97
NSHC41-45	0.78	2.25	5.08	0.78	2.33	5.45
Plain33	0.90	4.48	20.03	0.84	5.97	35.65
NSHC33-15	0.90	4.26	18.15	0.84	5.77	33.26
NSHC33-30	0.96	1.95	3.82	0.94	2.43	5.91
NSHC33-45	0.96	0.99	0.98	0.93	1.32	1.73
Plain28	0.82	5.53	30.59	0.95	3.05	9.32
NSHC28-15	0.94	3.13	9.82	0.96	2.51	6.28
NSHC28-30	0.87	4.66	21.74	0.90	4.14	17.14
NSHC28-45	0.60	6.94	48.17	0.68	6.45	41.64
NSHC41	0.89	3.40	11.55	0.82	4.47	20.00
NSHC33	0.89	4.55	20.72	0.81	6.12	37.40
NSHC28	0.87	5.03	25.26	0.88	4.76	22.65

## Data Availability

The data presented in this study are available from the corresponding author upon request.

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
