# Peer review of "Strength Prediction of Non-Sintered Hwangto-Substituted Concrete Using the Ultrasonic Velocity Method"

_materials, 2023, doi:10.3390/ma17010174_

Round 1

Reviewer 1 Report

Comments and Suggestions for Authors

This manuscript reported the properties of concrete with Kwangto as the substitute to cement and built a model that predict the strength of Kwangto blended concrete by the ultrasonic velocity method. The manuscript is overall well written, but there are still some points that need modification. Please refer to my comments below.

The Introduction part can be improved. In addition to seeking for cement substitutes without compromising performance, another important way to decarbonizing the cement industry is to use cement substitutes and balance their CO2 mitigation potential and performance. Please see C Li, et al., Cement and Concrete Composites 139 (2023) 105041.

Fig. 1. A scale bar is needed.

Lines 185 to 187. The discussion here is not clear. Please modify.

Section 3.1.2. The effect of NSH on the strength of concrete should be analyzed considering their pozzolanic activity. The optimized substitution level of supplementary cementitious materials, e.g., fly ash and slag, depends on the level of pozzolanic reactions. Please see J Li, et al., Resources, Conservation & Recycling 178 (2022) 106038.

Fig. 7. Scale bars are needed on these SEM images.

XRD peak assignments need to be justified. 1) CaO may not be found in hydrated cementitious materials. 2) C-S-H does not show sharp peaks. Instead, only a few broad peaks may be found in hydrated cementitious materials. 3) Please check whether there are peaks for unreacted cement clinkers, e.g., C3S and C2S. 4) I do not think it a good idea to use “Ca” for Ca(OH)2.

Table 7. Please point out what the table is referring to.

Comments on the Quality of English Language

English is overall fine.

Author Response

Comments: This manuscript reported the properties of concrete with Kwangto as the substitute to cement and built a model that predict the strength of Kwangto blended concrete by the ultrasonic velocity method. The manuscript is overall well written, but there are still some points that need modification. Please refer to my comments below.

Response: Thank you for taking your valuable time to review. We did our best to revise the paper by referring to the comments mentioned.

Comments 1: The Introduction part can be improved. In addition to seeking for cement substitutes without compromising performance, another important way to decarbonizing the cement industry is to use cement substitutes and balance their CO2 mitigation potential and performance. Please see C Li, et al., Cement and Concrete Composites 139 (2023) 105041.

Response 1: Agree. We added a new sentence to the "introduction" section referring to this paper.

Comments 2: Fig. 1. A scale bar is needed.

Response 2: Thank you for pointing this out. We add scale bar in the Fig. 1

Comments 3: Lines 185 to 187. The discussion here is not clear. Please modify.

Response 3: We agree with this comment. We add new sentence ‘The smaller the W/B of the concrete mix, the higher the compressive strength of the concrete, and as the NSH substitution ratio increased, the compressive strength decreased.’ after pointed out line to modify clearly.

Comments 4: Section 3.1.2. The effect of NSH on the strength of concrete should be analyzed considering their pozzolanic activity. The optimized substitution level of supplementary cementitious materials, e.g., fly ash and slag, depends on the level of pozzolanic reactions. Please see J Li, et al., Resources, Conservation & Recycling 178 (2022) 106038.

Response 4: Thank you for pointing this out. However, I would like to say this carefully: In the “introduction” section, NSH explains that pozzolanic reactions do not occur. Additionally, NSH felt that pozzolanic reactions did not occur until 7 days of age, making it difficult to mention them in section 3.1.2.

Comments 5: Fig. 7. Scale bars are needed on these SEM images.

Response 5: Thank you for pointing this out. We added scale bars in the figure. This SEM images were taken at 1500x magnification.

Comments 6: XRD peak assignments need to be justified. 1) CaO may not be found in hydrated cementitious materials. 2) C-S-H does not show sharp peaks. Instead, only a few broad peaks may be found in hydrated cementitious materials. 3) Please check whether there are peaks for unreacted cement clinkers, e.g., C3S and C2S. 4) I do not think it a good idea to use “Ca” for Ca(OH)2.

Response6: Thank you for your detailed teaching on XRD. We changed figure of XRD to fix those 3 problems in comment 6. And the 1), CaO was a mistake, so we delete CaO in the manuscript. Also to fix other problems, we changed figure, and it can describe more specific and easy to compare.

Comments 7: Table 7. Please point out what the table is referring to.

Response 7: Thank you for pointing this out. We added the explanation of table 7 to Section 3.2.2.

Reviewer 2 Report

Comments and Suggestions for Authors

Strength Prediction of Nonsintered Hwangto–Substituted Concrete Using Ultrasonic Velocity Method

In order for the work to be accepted, the following corrections must be made:

1.     Line 88: “Consequently, numerous researchers are employing UPV to evaluate the mechanical properties or quality of concrete.” For this sentence, a few references should be given from studies conducted in the last 5 years.

2.     Line 91 – 95: In the last paragraph, the difference of the study from other studies in the literature should be emphasized. What distinguishes this work from others?

3.     “Table 1. Experimental Plan” Here I recommend the authors to give the experimental plan as a workflow diagram instead of a table. In this way, the experiments will be easier to understand and the article will be richer in terms of visuality.

4.     Line 128: it should be “NHT: Non sintered Hwangto” > NSH

5.     I provide some new references, which may be included in the introduction section of the manuscript. “10.3390/polym15092127”, “10.3390/buildings13010045”.

6.     “3.1.1. Unit Weight” Why were the unit weights of samples containing 15% and 30% NSH not given in this section? It should be noted whether the unit weights were determined on samples cured for 28 days. Additionally, in this section, there is no discussion with the literature. A discussion should be added to this section.

7.     In Figure 3, standard error bars should be shown on the graphs. Also, the axis titles on the "x-axis" of the charts are misspelled. “x-axis” should not be displayed as Age (days).

8.     In Figure 4, standard error bars should be shown on the graphs.

9.     Table 6 is not referenced in the text. Additionally, it would be more appropriate to give this table in the "2.4. Test Method" section.

10.  What is the W/B ratio of the samples whose SEM images are presented? This value must also be specified in the text.

11.  Line 267: For “Calcium hydroxide”, its chemical designation should be written in parentheses.

12.  Table 7 is not referenced in the text.

13.  Model performance cannot be determined solely by the R2 value. For this, other performance analyzes such as RMSE, MAE, NS must also be calculated. In addition, the authors reported that exponential, linear and quadratic functions were used in the literature, and that the exponential function was suitable in this study. How did they decide this? My suggestion as a referee is this: prediction models should be created together with other functions (linear, quadratic, etc.), performance analyzes (R2, RMSE, MAE, NS) should be made and ultimately it should be proven that the exponential function is suitable. In its current state, the presented article is very weak in the prediction model section. In addition, the presented model was made with a very classical method. I think the authors should review the prediction model part and improve it further. For this, the authors can review the Conventional regression analysis (CRA) section of this article (https://doi.org/10.1016/j.conbuildmat.2022.129518).

14.  “A few suggestions for future studies can be given at the end of the "Conclusions" section.

15.  There is an error in the reference section; it should be rearranged according to the journal format.

Author Response

Comments: In order for the work to be accepted, the following corrections must be made:

Response: Thank you for taking your valuable time to review. We did our best to revise the paper by referring to the comments mentioned.

Comments 1: Line 88: “Consequently, numerous researchers are employing UPV to evaluate the mechanical properties or quality of concrete.” For this sentence, a few references should be given from studies conducted in the last 5 years.

Response 1: Thank you for pointing this out. We added three references.

Comments 2: Line 91 – 95: In the last paragraph, the difference of the study from other studies in the literature should be emphasized. What distinguishes this work from others?

Response 2: Thank you for pointing this out. We changed line 91 to 95 for explain more specific. ‘In this study, In order to measure the mechanical properties of concrete containing Hwangto, a natural mineral, the SEM and X-ray diffraction (XRD) analyses were performed by evaluating mechanical properties, such as unit weight, compressive strength, and stress–strain and its microstructure of NSHC concrete produced with various W/B and substitution ratios was measured.’.

Comments 3: “Table 1. Experimental Plan” Here I recommend the authors to give the experimental plan as a workflow diagram instead of a table. In this way, the experiments will be easier to understand and the article will be richer in terms of visuality.

Response 3: Agree. We change “Table 1. Experimental Plan” to “Figure 2. Workflow diagram of experimental plan” for make this article be richer in terms of visuality.

Comments 4: Line 128: it should be “NHT: Non sintered Hwangto” > NSH

Response 4: Thank you for pointing this out. We fixed NHT to NSH.

Comments 5: I provide some new references, which may be included in the introduction section of the manuscript. “10.3390/polym15092127”, “10.3390/buildings13010045”.

Response 5: Thank you for the new references. We added those references at 2nd paragraph in “introduction” section.

Comments 6: “3.1.1. Unit Weight” Why were the unit weights of samples containing 15% and 30% NSH not given in this section? It should be noted whether the unit weights were determined on samples cured for 28 days. Additionally, in this section, there is no discussion with the literature. A discussion should be added to this section.

Response 6:. We agree that must explain difference of unit weight. So, we added reason of unit weight difference at last 3.1.1 section. At first time, we thought that the unit weight was determined by the mixture used in the experiment. In addition, the difference in unit weight was judged to be due to a simple reason, referring to Table 4, so references were not included. Additionally, the difference in unit weight was not clear between 15% and 30%, so a comparison was made between Plain and 45%.

Comments 7: In Figure 3, standard error bars should be shown on the graphs. Also, the axis titles on the "x-axis" of the charts are misspelled. “x-axis” should not be displayed as Age (days).

Response 7: Thank you for pointing this out. We add standard error bars in Figure 3. Also we delete x-axis title because it doesn’t need in this figure.

Comments 8: In Figure 4, standard error bars should be shown on the graphs.

Response 8: Thank you for pointing this out. We add standard error bars in Figure 4 too.

Comments 9: Table 6 is not referenced in the text. Additionally, it would be more appropriate to give this table in the "2.4. Test Method" section.

Response 9: Agree, thank you for pointing this out. We added sentence in middle of “2.4. Test Method” section.

Comments 10: What is the W/B ratio of the samples whose SEM images are presented? This value must also be specified in the text.

Response 10: Thank you for pointing this out. Those SEM images are shows W/B 0.41 concrete, so we added W/B 0.41 in the “Figure 8” also start of the “3.2.1 SEM” section.

Comments 11: Line 267: For “Calcium hydroxide”, its chemical designation should be written in parentheses.

Response 11: Thank you for pointing this out. We added (Ca(OH)2) next to “Calcium hydroxide” to fix this problem.

Comments 12: Table 7 is not referenced in the text.

Response 12: Thank you for pointing this out. We added the explanation of table 7 to Section 3.2.2.

Comments 13: Model performance cannot be determined solely by the R2 value. For this, other performance analyzes such as RMSE, MAE, NS must also be calculated. In addition, the authors reported that exponential, linear and quadratic functions were used in the literature, and that the exponential function was suitable in this study. How did they decide this? My suggestion as a referee is this: prediction models should be created together with other functions (linear, quadratic, etc.), performance analyzes (R2, RMSE, MAE, NS) should be made and ultimately it should be proven that the exponential function is suitable. In its current state, the presented article is very weak in the prediction model section. In addition, the presented model was made with a very classical method. I think the authors should review the prediction model part and improve it further. For this, the authors can review the Conventional regression analysis (CRA) section of this article (https://doi.org/10.1016/j.conbuildmat.2022.129518).

Response 13: Thank you for pointing this out. Table 7 shows the process of finding a function that fits our data using the analysis you mentioned. According to R-square, Root-MSE and RMS, the exponential function shape was judged to be well suited except for some specimens (high-strength W/B range).

Comments 14: “A few suggestions for future studies can be given at the end of the "Conclusions" section.

Response 14: Agree. We added suggestions for future studies in the last of “Conclusion” section. Also, we will do more research about these to improve data of various concrete. Thank you.

Comments 15: There is an error in the reference section; it should be rearranged according to the journal format.

Response 15: Thank you for pointing this out. We fix reference and add DOI.

Reviewer 3 Report

Comments and Suggestions for Authors

The topic of the research work and manuscript is really interesting and provides new information. However there are some issues to be addressed towards its quality improvement before publication. You should provide more key words in order to ensure the detectability and readability of the article. In line 49, please provide the relevant study of https://doi.org/10.1016/j.jobe.2022.104913 as a reference to this statement. I believe that the paragraphs in lines 60-66 shouldd be removed and be placed on the methodolgoy chapter. In lines 75-76, the meaning is not very clear. Please, provide more images from the specimens preparation process. Please povide as well the standard deviation values on the bars of the graphs. Pay attention to  the superscripts/subscripts. The DOI numbers or links of the references in the end of manuscript are missing.  

Comments on the Quality of English Language

The use of English language is in general acceptable.

Author Response

Comments: The topic of the research work and manuscript is really interesting and provides new information. However there are some issues to be addressed towards its quality improvement before publication.

Response: Thank you for taking your valuable time to review. We did our best to revise the paper by referring to the comments mentioned.

Comments 1: You should provide more key words in order to ensure the detectability and readability of the article.

Response 1: Thank you for pointing this out. We add 2 keywords “regression analysis”, “microstructural analysis”.

Comments 2: In line 49, please provide the relevant study of https://doi.org/10.1016/j.jobe.2022.104913 as a reference to this statement.

Response 2: Thank you for recommending reference for us. We added this reference at “Introduction” section.

Comments 3: I believe that the paragraphs in lines 60-66 should be removed and be placed on the methodolgoy chapter.

Response 3: Thank you for pointing this out. We replaced lines 60-66 to section “2.2 Materials”.

Comments 4: In lines 75-76, the meaning is not very clear.

Response 4: Agree. We change lines 75-76 to “Concrete using NSH cannot be expected to develop strength due to pozzolanic reaction in the short or long term.” For make sentence has clear meaning.

Comments 5: Please, provide more images from the specimens preparation process.

Response 5: We put new figure “NSHC preparation overview” end of section “2.1 experimental plan”. This can be make this paper richer visuality, thank you.

Comments 6: Please provide as well the standard deviation values on the bars of the graphs.

Response 6: Thank you for pointing this out. We add standard error bars in Figure 3 and Figure 4.

Comments 7: Pay attention to the superscripts/subscripts.

Response 7: Thank you for pointing this out. We fix subscripts in Figure 8. We will careful to use superscripts and subscripts.

Comments 8: The DOI numbers or links of the references in the end of manuscript are missing.

Response 8: Thank you for pointing this out. We added DOI in the end of manuscript.

Reviewer 4 Report

Comments and Suggestions for Authors

In this work, the authors prepared and characterized concrete with a content of nonsintered Hwangto at different compositions. The manuscript is clear and detailed, the experiments and the results are well described, and the conclusions are following results, therefore, it can be considered for publication if the following revision is considered:

-        The maximum value of strain in the scales of Figures 6 should be changed to 0.003 to better distinguish the different curves.

-        Please increase the quality or size or color of figures 9 to better distinguish between the different experimental points.  The same for figures 10.

-        Please explain better the models used in section 3.3, namely the ones presented in figures in figure 10. Please indicate in the legend of figure 10 on the right the meaning of the lines represented. Distinguish the 2 graphs in figure 10 in the text and legend.

-        It should be made clear in the discussion and conclusions the main advantages of this analyzed material in comparison with others in the literature. A table with this comparison is welcome.

Author Response

Comments: In this work, the authors prepared and characterized concrete with a content of nonsintered Hwangto at different compositions. The manuscript is clear and detailed, the experiments and the results are well described, and the conclusions are following results, therefore, it can be considered for publication if the following revision is considered:

Response: Thank you for taking your valuable time to review.

Comments 1: The maximum value of strain in the scales of Figures 6 should be changed to 0.003 to better distinguish the different curves.

Response 1: Thank you for pointing this out. We fixed maximum strain scales 0.0055 to 0.003 for better distinguish the different curves.

Comments 2: Please increase the quality or size or color of figures 9 to better distinguish between the different experimental points. The same for figures 10.

Response 2: Thank you for pointing this out. We change size of figures bigger for observed well. Also we changed figure not only 9, 10 but also 3, 4, 5, 6.

Comments 3: Please explain better the models used in section 3.3, namely the ones presented in figures in figure 10. Please indicate in the legend of figure 10 on the right the meaning of the lines represented. Distinguish the 2 graphs in figure 10 in the text and legend.

Response 3: Thank you for pointing these out. We added more explain of legend. And next we put new legend at figure 10 (b) to represent meaning of the lines. Also, we added “(a) comparison with previous research” and “(b) comparison between experimental and predicted Fc” for distinguish the figures.

Comments 4: It should be made clear in the discussion and conclusions the main advantages of this analyzed material in comparison with others in the literature. A table with this comparison is welcome.

Response 4: Thanks for the good point. For concrete using other existing cement substitutes, formulas have been prepared to predict strength, or numerous studies have been conducted on NSH comparisons. In this paper, we present our own strength prediction model for concrete using NSH and do our best to compare it with existing papers and standards. I would like to say that this is the main strength of our paper.

Round 2

Reviewer 2 Report

Comments and Suggestions for Authors

The authors have made the necessary changes. Therefore, the manuscript can be accepted.

Reviewer 3 Report

Comments and Suggestions for Authors

As I have checked the authors have implemented the proposed changes in the revised verion of manuscript towards the improvement of their work. Almost all the changes have been implemented and in my opinion, the manuscript is well-prepared and organized enough to be accepted for publication in this journal.

Comments on the Quality of English Language

The English language use has been improved.

Reviewer 4 Report

Comments and Suggestions for Authors

This manuscript has been improved considering the reviewers' suggestions and, therefore, I can recommend this work for publication